# Effect of Erector Spinae Plane Block on Postoperative Pain after Laparoscopic Colorectal Surgery: A Randomized Controlled Study

**DOI:** 10.3390/jpm12101717

**Published:** 2022-10-14

**Authors:** Jung Ju Choi, Young Jin Chang, Dongchul Lee, Hye Won Kim, Hyun Jeong Kwak

**Affiliations:** Department of Anesthesiology and Pain Medicine, Gachon University Gil Medical Center, 783 Namdong-daero, Namdong-gu, Incheon 21556, Korea

**Keywords:** erector spinae plane block, laparoscopic colorectal operation, postoperative pain

## Abstract

The erector spinae plane (ESP) block can be used to reduce pain and opioid requirements after abdominal surgery. We evaluated the effect of the ESP block on postoperative pain score, analgesic use, and quality of recovery (QoR) score in patients undergoing laparoscopy. Fifty-nine patients undergoing elective laparoscopic colorectal surgery were randomly assigned to control (*n* = 30) or ESPB (*n* = 29) groups after anesthesia induction. In the ESPB group, an ultrasound-guided ESP block was performed immediately after induction using 20 mL of 0.5% ropivacaine bilaterally. The primary outcome was the postoperative pain score, which was evaluated using the 11-point numeric rating scale (NRS) (0 = no pain, 10 = worst imaginable pain), in the recovery room. NRS “at rest” and “on cough” and total dose of fentanyl rescue (in the recovery room) as well as NRS “at rest” and the cumulative administered fentanyl dose of patient-controlled analgesia (24 h post-surgery) were significantly lower in the ESPB group than in the control group. The postoperative QoR score did not differ between the groups. Bilateral ESP block after induction reduced pain scores and opioid requirements for 24 h postoperatively but did not improve the QoR in patients undergoing laparoscopic colorectal surgery.

## 1. Introduction

In colorectal surgery, conventional laparotomy has recently been replaced by laparoscopic surgery, which provides reduced postoperative pain and postoperative stress response [1,2,3,4]. After elective colorectal surgery, more than half of patients experience moderate to severe pain regardless of the surgical method. In general, laparoscopic colorectal surgery is less painful than open laparotomy. However, an earlier report has shown that laparoscopic surgery could be more painful on the first postoperative day [5]. Therefore, aggressive pain management is required even after laparoscopic colorectal surgery, particularly up to 24 h immediately after surgery [5].

The erector spinae plane (ESP) block is a recently introduced interfascial plane block for the control of thoracic neuropathic pain [6,7], which provides postoperative analgesia after breast, thoracic, and abdominal surgeries [6,7,8]. The ESP block is a simpler and safer procedure compared to epidural and paravertebral blocks because it has an easily recognizable sonoanatomy and no structures nearby pose a risk of needle injury. In addition, the ESP block does not have the risks of developing hypotension associated with epidural analgesia and epidural spread or vascular puncture associated with paravertebral blocks.

We hypothesized that an ultrasound (US)-guided ESP block immediately after induction might decrease postoperative pain score and analgesic requirements and, thus, improve the quality of recovery (QoR) after laparoscopy. Therefore, we investigated whether Bilateral ESP blockade after induction reduced postoperative pain score and opioid requirements and improved the QoR in patients undergoing laparoscopic colorectal surgery.

## 2. Materials and Methods

This study was conducted after receiving approval from the ethics committee of Gachon University Gil Hospital (GFIRB2020-001) and registration at www.ClinicalTrials.gov (NCT04238780), accessed on 14 October 2020. All enrolled patients provided informed consent before undergoing the operation. We recruited patients aged 20–70 years who were scheduled to undergo elective laparoscopic colorectal surgery due to malignant disease. We excluded patients with an American Society of Anesthesiologists physical status ≥ 3, with a body mass index > 35 kg/m^2^, receiving anticoagulant therapy, with bleeding disorders, with hypersensitivity to local anesthetics, with spine or chest wall deformity, pregnancy, and with tolerance to opioid analgesics. Using a computer-generated randomization protocol, we randomly assigned the patients to the ESPB (*n* = 30) or control (*n* = 30) groups. After obtaining informed consent the day before surgery, the preoperative QoR was evaluated using a 40-item questionnaire to access five recovery domains (QoR-40) [9].

Routine monitoring was performed in the operating room, including non-invasive blood pressure, electrocardiogram, pulse oximetry, end-tidal carbon dioxide, and bispectral index (BIS Vista Monitor, revision 3.0; Aspect Medical Systems, Norwood, MA, USA). Anesthesia induction was performed with lidocaine (0.5–1.0 mg/kg), remifentanil (0.5–1.0 μg/kg), propofol (1.5–2.0 mg/kg), and rocuronium (0.6–0.8 mg/kg), and was maintained with sevoflurane (2–2.5 vol%) and remifentanil infusion (0.05–0.15 μg/kg/min).

In the ESPB group, after anesthesia induction, an ESP block was performed by one designated anesthesiologist. After laying the patient on his or her side, a convex US probe was placed 2–3 cm lateral to the spine. The transverse process of T7 was identified using ultrasonic waves, skin, trapezius, and visible effector spinae muscle. Under the ultrasonic guidance, a 22-G block needle was inserted in the cranial-to-caudal direction in-plane and brought into contact for injection near the T7 transverse process. After confirming the needle location, 0.5–1 mL of saline was injected to confirm the location; then, 20 mL of 0.5% ropivacaine was administered on each side. Ten minutes after ESP block, another anesthesiologist, blinded to the patient blocking, maintained the patient’s anesthesia and performed the postoperative management and evaluations.

All patients received patient-controlled analgesia (PCA) using an infuser (Accufuser Plus^®^, Wooyoung Medical, Seoul, Korea) for 48 h after surgery. The PCA contained fentanyl (800 μg) in normal saline (100 mL) and was administered at a basal infusion rate of 2 mL/h and an intermittent bolus of 0.5 mL with a lock-out interval of 15 min. To prevent postoperative nausea and vomiting (PONV), 0.3 mg intravenous ramosetron was administered before surgery completion.

The pain scores were evaluated using the 11-point numeric rating scale (NRS) (0 = no pain, 10 = worst pain imaginable) measured after arrival in the recovery room. If NRS ≥ 4 or higher in the recovery room, 50 μg fentanyl was administered up to six times. In the ward, the pain scores, total cumulative fentanyl dose of PCA, and rescue analgesic dose were recorded at 6 and 24 h after surgery. If the NRS was >4 or upon patient request in the ward, tramadol (100 mg), ketorolac (30 mg) or ketoprofen (100 mg) was administered intravenously. The analgesic type and dose were determined by an attending surgeon. Postoperative analgesic consumption in the ward were converted to intravenous morphine equivalents and recorded. Previous study reported that intravenous tramadol has the same analgesic potency as one-tenth that of morphine [10]. Furthermore, previous studies reported that ketorolac 30 mg or ketoprofen 50 mg had an equivalent effect as morphine 12 mg [11,12,13]. Thus, we converted tramadol 100 mg, ketorolac 30 mg and ketoprofen 100 mg to morphine 10 mg, 12 mg, and 24 mg, respectively. Postoperative QoR was assessed based on the QoR-40 performed 24 h after surgery.

The primary study outcome was the postoperative pain score in the recovery room. Since the previous study reported that the intensity of postoperative pain was the most severe in the recovery room [14], we thought that the control of pain in the recovery room is very important. Thus, we set postoperative pain score in the recovery room as the primary outcome to check the effect of the ESP block. The secondary outcomes were requirements for rescue analgesia and QoR-40 score. A previous study of laparoscopic colorectal surgery reported a mean (standard deviation, SD) postoperative pain score of 5.9 (2.0) in the recovery room [14]. We assumed a mean difference in pain score between the control and ESPB groups of 30%. Thus, 27 patients were required per group, assuming an α-error of 0.05 and a β-error of 0.1. We included 30 patients in each group due to the possibility of drop-outs.

Data were analyzed using IBM SPSS Statistics for Windows, version 19.0 (IBM Corp., Armonk, NY, USA). Variables were shown as means (SD), medians (interquartile ranges, [IQR]), or number of patients. The normality of continuous variables was assessed by Kolmogorov–Smirnov tests. Continuous variables were compared using independent t-tests for normally distributed data or Mann–Whitney U tests for skewed data. Categorical data were analyzed by χ^2^ or Fisher’s exact tests, as appropriate. *p* < 0.05 was considered statistically significant.

## 3. Results

Although 60 patients were enrolled, one patient in the ESPB group who refused to fill out a postoperative QoR-40 was excluded from the final analysis; thus, 59 patients were analyzed (Figure 1). The perioperative data and patient characteristics, which did not differ between the groups, are listed in Table 1.

Figure 2 illustrated pain scores in the recovery room, and at 6 h and 24 h after surgery. In the ESPB group vs. the control, NRS (medians [IQR]) at rest in the recovery room (4 [3–5] vs. 5 [4–7], *p* = 0.009), at 6 h after surgery (2 [1–3] vs. 3 [2–4], *p* = 0.003) and at 24 h after surgery (2 [1–2] vs. 2 [2–3], *p* = 0.002), were significantly low. In the ESPB group vs. the control, NRS on cough in the recovery room (5 [4–6] vs. 6 [5–8], *p* = 0.001) and at 6 h after surgery (3 [3–4] vs. 4 [3–5], *p* = 0.040) were significantly low, while the NRS on cough at 24 h after surgery did not differ between the groups (3 [3–4] vs. 3 [2–4], *p* = 0.158).

In the recovery, the total administered dose of fentanyl bolus (50 [50–100] μg, *p* = 0.032) were significantly lower in the ESPB group than in the control group. The number of patients requiring fentanyl bolus did not differ significantly between the groups (27 [90%] vs. 24 [83%], *p* = 0.472). At 6 h after surgery, the cumulative administered fentanyl dose of PCA (160 [160–180] μg vs. 200 [160–240] μg, *p* = 0.045) was significantly lower in the ESPB group than in the control group. The numbers of patients requiring analgesic rescue and the morphine equivalents of additional analgesics did not differ significantly between the groups. The cumulative administered fentanyl dose of PCA was significantly lower in the ESPB group than in the control group (480 [400–560] μg vs. 560 [480–600] μg, *p* = 0.031). The numbers of patients requiring rescue analgesics and the morphine equivalents of additional analgesics did not differ significantly between the control and the ESPB groups (Table 2).

The global QoR-40 score did not differ significantly between the groups preoperatively and postoperatively. The dimension scores for emotional state, physical comfort, psychological support, physical independence, and pain did not significantly differ between groups (Table 3).

## 4. Discussion

The results of this study showed that bilateral ESP block after induction significantly reduced the postoperative pain score and opioid requirements but did not improve the QoR after laparoscopic colorectal surgery. This study was the first randomized controlled trial to examine the effect of ESP block on pain, opioid requirements, and QoR after laparoscopic abdominal surgery.

Although the intensity of pain after laparoscopic surgery is less than that of laparotomy, it is not pain-free. Pain after laparoscopic surgery is related to nerve traction, blood vessel damage, and the release of inflammatory substances due to peritoneal distension [5]. A previous study has shown that laparoscopic surgery could show high intensity of pain and high requirements of analgesics immediately after surgery [5]. This is consistent with our results showing that the control group had a median NRS > 4 (5 at rest and 6 on cough) and that 90% of patients required rescue fentanyl in the recovery room.

US-guided ESP block was first introduced for the management of acute and chronic chest pain [15,16]. ESP block involves the spread of a local anesthetic into the paravertebral space, which is effective for the management of somatic and visceral pain. This effect is similar to that of epidural block [6]. Moreover, because the ESP block is performed under US guidance far from the spinal cord, the risk of complications such as spinal cord injury, hematoma, and pneumothorax is low. Recent studies have reported that ESP block at the thoracic level effectively reduced analgesic requirements and relieved pain after abdominal hysterectomy, sleeve gastrectomy, lumbosacral spine surgery, and laparoscopic cholecystectomy [16,17,18,19,20,21,22]. Patients who underwent ESP blockade at the T7 level had sensory blockade from T6 to T12; thus, the ESP block can be performed at the T7 or T8 levels for postoperative pain management after abdominal surgery [18]. The present study performed the ESP blockade at the T7 level under US guidance, with no complications associated with the ESP blockade.

Pain after laparoscopic colorectal surgery originates from the abdominal wall incision and visceral dissection. However, the regional blocks including the ESP block may improve the parietal pain but not the visceral pain [23]. Intraperitoneal instillation of ropivacaine has shown great efficacy in the treatment of pain, opioid consumption and postoperative recovery in several abdominal surgeries including colorectal surgery [24]. In this study, the pain scores of ESPB group, despite the significant statistical difference, remained high (median NRS at rest was 4 and NRS on cough was 5 in the recovery room). This result is most likely due to the visceral pain that remains undertreated.

According to the Enhanced Recovery After Surgery Society (ERAS) recommendations, proper management of pain in patients undergoing elective colorectal surgery is vital for recovery [25]. Avoiding the use of opioids, regardless of laparoscopic or open surgical method, is associated with a faster recovery of bowel function and early mobilization. As reported, opioid overuse causes difficulty in early mobilization and is associated with prolonged hospitalization and increased hospital readmission rates [25]. The ERAS guidelines recommend epidural analgesia or transverse abdominis plane (TAP) block to avoid opioid use and apply multimodal analgesia [25]. A study of total abdominal hysterectomy showed that the ESP block provided longer and more potent postoperative analgesia with significantly less morphine consumption compared to the TAP block [18]. Moreover, the ESP block consistently reduced pain scores and opioid requirements from the recovery room to 24 h after surgery. Thus, the ESP block might be an option for multimodal analgesia as an alternative to epidural or TAP blocks in colorectal surgery. We assumed that the ESP block might have improved the QoR score after surgery as the ESP block has opioid-sparing and analgesic effects. However, the global and individual dimensions of the QoR score did not differ significantly in the 24 h after surgery. The reason for these results may be that the opioid-sparing effects of ESP block, such as the rapid recovery of bowel function and early mobilization, might appear only 24 h after surgery. Further studies on the effects of ESP block on the recovery of bowel function or the length of hospital stay might be needed. In this study, we used a 40-item QoR-40 score instead of a short-form 15-item postoperative QoR score (QoR-15), because QoR-40 can provide a more extensive evaluation of a patient’s QoR. However, since all dimension scores of QoR-40 were comparable between the two groups in this study, we think that the adoption of QoR-15 would not have affected the results on pain control.

Our study has several limitations. First, because the ESP block was performed after anesthesia induction while the patient was unconscious, we could not confirm the blocked level or blockade strength. However, we confirmed the spread of the local anesthetic by ultrasound, and no patients developed side effects such as pneumothorax. Second, in this study, the placebo injection was not performed in the control group. In a randomized controlled study, a placebo injection might be needed, but since the side effects such as pneumothorax and nerve injury related to the placebo injection cannot be completely excluded, the control group did not receive a placebo injection in this study for the safety of patient. Furthermore, ESP block was not compared to other regional anesthetic techniques such as epidural analgesia or TAP block, which are also effective for the management of postoperative pain after laparoscopic surgery. Therefore, future studies comparing ESP block to other regional blocks might be required.

In conclusion, bilateral ESP blockade after induction reduced postoperative pain and opioid requirements in patients undergoing laparoscopic colorectal surgery. However, the QoR was not improved on the first postoperative day.

## Figures and Tables

**Figure 1 jpm-12-01717-f001:**
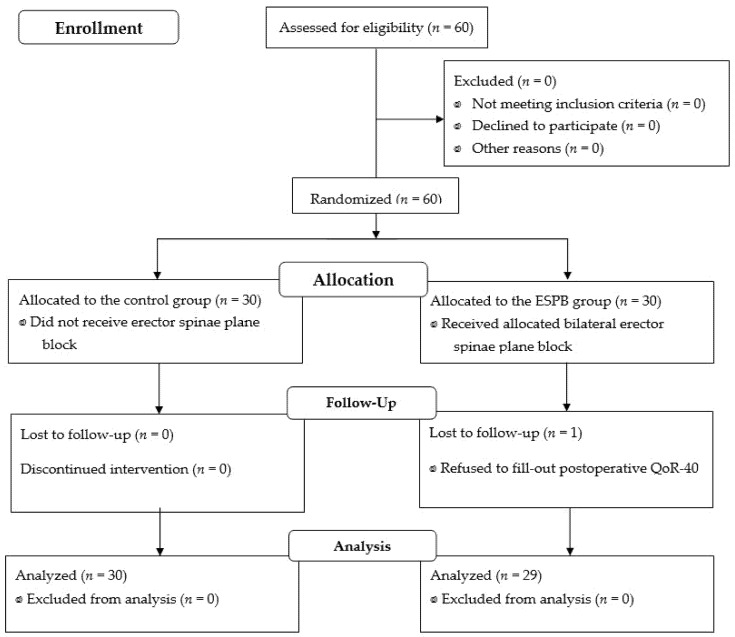
Flow diagram of patient’s allocation.

**Figure 2 jpm-12-01717-f002:**
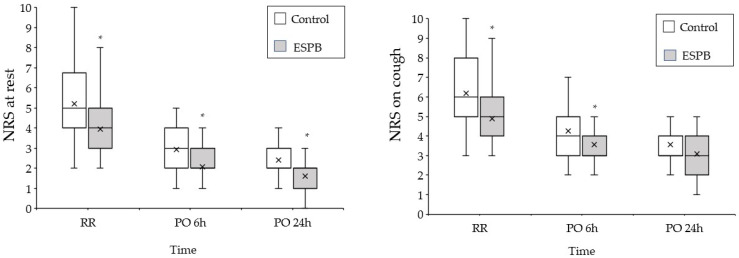
Pain scores at rest and on cough in the recovery room, and at 6 h and 24 h after the surgery. Control group: patients without nerve block, ESPB group: patients with bilateral single erector spinae plane block. ESPB: erector spinae plane block, NRS: 11-point numeric rating score from 0 (no pain) to 10 (worst pain imaginable). RR: recovery room, PO 6 h and 24 h: 6 h and 24 h after the surgery, respectively. Box, the line in the box, X and error bar mean interquatile range, median, mean and maximum or minimum value, respectively. *: *p*-value < 0.05 between two groups.

**Table 1 jpm-12-01717-t001:** Patient characteristics and perioperative clinical data.

	Control (*n* = 30)	ESPB (*n* = 29)	*p*-Value
Age (y)_	59.5 (7.9)	58.6 (7.1)	0.618
Sex (M/F)	30 (15/15)	29 (17/12)	0.604
Weight (kg)	63.3 (10.8)	65.6 (12.5)	0.452
Height (cm)	162 (9)	162 (11)	0.772
Surgery duration (min)	143 (49)	144 (49)	0.935
Anesthesia duration (min)	189 (55)	195 (51)	0.691
Intraoperative remifentanil (μg)	630 (100–1720 [484–1000])	600 (140–1086 [400–760])	0.358

Values are mean (standard deviation) or number of patients or median (range [interquartile range]). Control group: patients without nerve block, ESPB group: patients with bilateral single erector spinae block. ESPB: erector spinae plane block.

**Table 2 jpm-12-01717-t002:** Total administered dose of fentanyl bolus or cumulative administered fentanyl dose of PCA, and number of patients who required rescue analgesics in the recovery room, and at 6 h and 24 h after surgery.

Variables	Control (*n* = 30)	ESPB (*n* = 29)	*p*-Value
**Recovery room**			
Total administered dose of fentanyl bolus (μg)	100 (0–300 [50–100])	50 (0–200 [50–100])	0.032
Fentanyl rescue bolus	27 (90%)	24 (83%)	0.472
**6 h after surgery**			
PCA fentanyl dose (μg)	200 (160–240 [80–320])	160 (160–180 [40–240])	0.045
Rescue analgesics	20 (67%)	16 (55%)	0.365
Morphine equivalents (mg)	10 (0–34 [0–24])	10 (0–24 [0–24])	0.515
**24 h after surgery**			
PCA fentanyl dose (μg)	560 (376–800 [480–600])	480 (80–720 [400–560])	0.031
Rescue analgesics	13 (43%)	10 (34%)	0.486
Morphine equivalents (mg)	0 (0–58 [0–12.5])	0 (0–20 [0–10])	0.466

Values are median (range [interquatile range]) or number of patients (%). Control group: patients without nerve block, ESPB group: patients with bilateral single erector spinae plane block. PCA: patient-controlled analgesia, NRS: 11-point numeric rating score. Morphine equivalents: the consumption of other types of postoperative opioids was converted to intravenous morphine equivalents. PCA: patient-controlled analgesia including fentanyl (800 μg) in normal saline (100 mL) (basal infusion rate of 2 mL/h, intermittent bolus of 0.5 mL with a lock-out in-terval of 15 min).

**Table 3 jpm-12-01717-t003:** Quality of recovery scales.

	Control (*n* = 30)	ESPB (*n* = 29)	*p*-Value
**Preoperative QoR score**			
Total score	182 (17)	182 (17)	0.535
Emotional state	40 (7)	40 (5)	0.839
Physical discomfort	55 (4)	55 (5)	0.914
Psychological support	32(4)	32 (5)	0.657
Physical independence	22 (4)	23 (4)	0.969
Pain	34 (2)	33 (2)	0.284
**Postoperative QoR score**			
Total score	163 (24)	166 (22)	0.922
Emotional state	37 (7)	38 (6)	0.664
Physical discomfort	50 (6)	51 (6)	0.266
Psychological support	30(5)	31 (6)	0.782
Physical independence	16 (7)	16 (7)	0.827
Pain	30 (4)	30 (4)	0.797

Values are mean (standard deviation). Control group: patients without nerve block, ESPB group: patients with bilateral single erector spinae plane block. QoR: quality of recovery, ESPB: erector spinae plane block.

## Data Availability

Data is contained within the article.

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
