# Peer review of "Effect of Erector Spinae Plane Block on Postoperative Pain after Laparoscopic Colorectal Surgery: A Randomized Controlled Study"

_jpm, 2022, doi:10.3390/jpm12101717_

Round 1

Reviewer 1 Report

This study showed that bilateral ESP block significantly reduced the postoperative pain score and opioid requirements but did not improve the QoR after laparoscopic colorectal surgery. This topic has the clinical significance. However, I have some concerns about the clinical trial design and the results. 

1 Line 17, Line 163,   the ESP block was preoperatively performed.  Line 208, the procedure was performed after anesthesia induction.    Please clearly stated the precise time.  The current description may lead to the misunderstanding. 

2 Line 90 . Different types of postoperative analgesic consumption were converted to intravenous morphine equivalent.    Please introduce the detailed method in the manuscript.

3 Line 94 The primary study outcome was the postoperative pain score in the recovery room.  Why did the authors choose the  postoperative pain score in the recovery room as the primary study outcome? Please clearly state.

4 Line 95-98 A previous study of laparoscopic colorectal surgery reported a mean (standard deviation, SD) postoperative pain score of 5.9 (2.0) in the recovery room. We assumed a mean difference in pain score between the control and ESPB groups of 20%.   The authors  assumed a mean difference in pain score between the control and ESPB groups of 20%. It was 4.72, which is more than 4.  However, NRS should be controlled < 4 under a standardized analgesic regimen.   

5 Line 121 In the recovery room, NRS at rest (4 [3–5] vs. 5 [4–7], p = 0.009), NRS on cough (5 [4– 6] vs. 6 [5–8], p = 0.001), and the total dose of fentanyl rescue (100 [50–100] μg vs. 50 [50– 100] μg, p = 0.032) were significantly low in the ESPB group than in the control group.    The mistake should be corrected.

6 In the recovery room, NRS at rest was 5 [4-7] in the control group. It indicated that many patients had the NRS more than 4.   It may be attributed to the unsuccessful analgesic regimen in this study. 

Author Response

Dear reviewer #1,

Thank you for reviewing the manuscript.  Answers to your suggestions are written below and changed parts of the manuscript are written in red characters.

Response to Reviewer 1 Comments

Point 1: Line 17, Line 163,   the ESP block was preoperatively performed.  Line 208, the procedure was performed after anesthesia induction.    Please clearly stated the precise time.  The current description may lead to the misunderstanding.

Thank you for your advice. We revised the sentences as belows.

Abstract :”preoperatively” à “immediately after induction “

         “Preoperative bilateral ESP block” à “Bilateral ESP block after induction “

Introduction :

“ preoperative ultrasound (US)-guided ESP block “

 “an ultrasound (US)-guided ESP block immediately after induction”

  Discussion : “preoperative bilateral ESP block” à “bilateral ESP block after induction”

Point 2: Line 90 . Different types of postoperative analgesic consumption were converted to intravenous morphine equivalent.    Please introduce the detailed method in the manuscript.

Thank you for your advice. We added the description of how different types of postoperative analgesic consumption were converted to intravenous morphine equivalent in the methods section.

“If the NRS was >4 or upon patient request in the ward, tramadol (100 mg), or ketoprofen (100 mg) was administered intravenously. The analgesic type and dose were determined by an attending surgeon. Postoperative analgesic consumption in the ward were converted to intravenous morphine equivalents and recorded. Previous study reported that intravenous tramadol has the same analgesic potency as one-tenth that of morphine [10]. And previous studies reported that ketolorac 30 mg or ketoprofen 50 mg had an equivalent effect as morphine 12 mg [11-13]. Thus, we converted tramadol 100 mg, ketorolac 30 mg and ketoprofen 100 mg to mophine 10 mg, 12 mg, and 24 mg, respectively.”

Point 3: Line 94 The primary study outcome was the postoperative pain score in the recovery room.  Why did the authors choose the  postoperative pain score in the recovery room as the primary study outcome? Please clearly state.

Since the previous study reported that the intensity of postoperative pain was the most severe in the recovery room [14], we thought that the control of pain in the recovery room is very important. Thus, we set pain score in the recovery room as the primary outcome to check the effect of the ESP block. We added these comments in the method section.

        “Since the previous study reported that the intensity of postoperative pain was the most severe in the recovery room [14], we thought that the control of pain in the recovery room is very important. Thus, we set postoperative pain score in the recovery room as the primary outcome to check the effect of the ESP block.”

Point 4: Line 95-98 A previous study of laparoscopic colorectal surgery reported a mean (standard deviation, SD) postoperative pain score of 5.9 (2.0) in the recovery room. We assumed a mean difference in pain score between the control and ESPB groups of 20%.   The authors  assumed a mean difference in pain score between the control and ESPB groups of 20%. It was 4.72, which is more than 4.  However, NRS should be controlled < 4 under a standardized analgesic regimen.

We agree with you. As you pointed out, pain with NRS > 4 should be controlled. However, we did not assume that preoperative bilateral ESPB alone could reduce pain scores to the extent that there is no additional analgesic drugs in the recovery room. We anticipated that ESPB could reduce not only the use of analgesic rescue, but also adverse events related with analgesics, such as nausea, respiratory depression, and delayed recovery.

Also, We apologized the mistake. In this study, assumed mean difference between two groups iss not 20%, but 30% (5.9 vs. 4.13). We corrected it in the method section.

We assumed a mean difference in pain score between the control and ESPB groups of 30%.”

Point 5: Line 121 In the recovery room, NRS at rest (4 [3–5] vs. 5 [4–7], p = 0.009), NRS on cough (5 [4– 6] vs. 6 [5–8], p = 0.001), and the total dose of fentanyl rescue (100 [50–100] μg vs. 50 [50– 100] μg, p = 0.032) were significantly low in the ESPB group than in the control group.    The mistake should be corrected.

Thank you for your advice. We corrected them.

“In the recovery, the total administered dose of fentanyl bolus (50 [50–100] μg vs. 100 [50–100] μg, p = 0.032) were significantly lower in the ESPB group than in the control group.”

Point.6: In the recovery room, NRS at rest was 5 [4-7] in the control group. It indicated that many patients had the NRS more than 4.   It may be attributed to the unsuccessful analgesic regimen in this study. 

  • We agree with you. As you pointed out, according to our analgesic regimen, the pain intensity should be controlled NRS 4 or less. However, previous study also reported that postoperative pain at 2 hours after laparoscopic colorectal surgery exceeded VAS 50mm, despite the administration of rescue analgesics when the pain was VAS 30mm or higher [3]. And, because we should administer rescue analgisics considering about adverse events related with analgesics, such as nausea, respiratory depression, and delayed recovery, sufficient analgesia could not be obtained with rescue analgesics alone.

Reviewer 2 Report

Review of JPM 1926767

In this article, Choi et al report a randomized control trial of a relatively new anesthesia technique, erector spinae plane (ESP) block, for patients undergoing laparoscopic colorectal surgery. This study was a well designed randomized trial in an identified area of need with a novel and promising anesthesia technique and provides valuable information for the ongoing management of colorectal surgery patients. ESP block is associated with minimal side effects, has been reported to extend to the thoracic and abdominal areas and may provide advantages over other anesthesia techniques. In addition, post-operative pain after laparoscopic colorectal surgery was an identified area with currently inadequate pain management. The authors thus designed this study to investigate ESP as an option for these patients. Their reported results show promise that this technique may be beneficial for laparoscopic colorectal surgeries. While the study would be a valuable addition to the field, it requires revision to clarify details and further discussion before considering publication.

Major comments:

1) The control group in this study received no additional anesthetic intervention and no placebo injection. Given there may be risks of a placebo injection as a control for ESP block there may have been a decision to have no intervention in control which is reasonable be should be explicitly discussed in the manuscript. The possible non-specific effects or risks of the intervention vs no intervention controlled trial should be discussed and recognized as a limitation of the study.

2) The authors should provide a clear explanation of why they considered the recovery room pain scores the primary end point of the study. Is this related to the expected duration of effect of the ESP block? If this when pain scores are noted to be highest? This would be helpful to understand the objectives of the study and how to apply the use of this technique to other areas of pain management.

Minor comments:

3) Results – Page 4 lines 121-123: In the results section where the authors describe their primary results the sentence is difficult to follow. The results are listed in the text for multiple parameters sequentially; however, the corresponding patient groups are not listed until after the data. Thus it is unclear which data point refers to which group until reaching the end of the sentence. The sentence should be re-structured so the reader can more easily follow these important and primary results of the study. List the two groups before the data. Ie. “In ESPB group vs control, NRS at rest (4 vs 5) …”etc

4) For the recovery room timepoint (Table 2) Clarify if “total dose of fentanyl rescue” (used in text) and “Total administered dose of fentanyl” (used in table) are only bolus doses vs including the basal rate. Use a consistent terminology between these two terms uses in text and in the figure.

5) Since Table 2 and 3 are reporting similar data at separate time points they could be grouped into one table. If this was decided to keep recovery room separate from other time points the importance of this may be clarified by the authors response to point 2.

6) In 6 hour and 24 hour timepoint, (Table 3) – Please clarify again whether cumulative fentanyl dose includes basal infusion and boluses.

7) Since medians are reported for NRS it makes it difficult to appreciate the difference in the groups. For 24 hour timepoint median is reported as 2 for both groups with a significant difference of p= 0.002. The authors might consider a visual representation of the difference between the groups for the most important time points. For example a dot plot where you can see the distribution of pain scores for the ESP block group vs control? This may add a nice visual to support the benefit and could also display the change over the first 24 hours for each group if done for recovery room, 6 hour and 24 hours time points.

Author Response

Dear reviewer #2,

Thank you for reviewing the manuscript.  Answers to your suggestions are written below and changed parts of the manuscript are written in red characters.

Response to Reviewer 2 Comments

Point 1: The control group in this study received no additional anesthetic intervention and no placebo injection. Given there may be risks of a placebo injection as a control for ESP block there may have been a decision to have no intervention in control which is reasonable be should be explicitly discussed in the manuscript. The possible non-specific effects or risks of the intervention vs no intervention controlled trial should be discussed and recognized as a limitation of the study.

We agree with you. We added the comments about this as a limitation in the discussion section.

       “ Second, in this study, the placebo injection was not performed in the control group. In a randomized controlled study, a placebo injection might be needed, but since the side effects such as pneumothorax and nerve injury related to the placebo injection cannot be completely excluded, the control group did not receive a placebo injection in this study for the safety of patient.”

Point 2: The authors should provide a clear explanation of why they considered the recovery room pain scores the primary end point of the study. Is this related to the expected duration of effect of the ESP block? If this when pain scores are noted to be highest? This would be helpful to understand the objectives of the study and how to apply the use of this technique to other areas of pain management.

Since the previous study reported that the intensity of postoperative pain was the most severe in the recovery room [14], we thought that the control of pain in the recovery room is very important. Thus, we set pain score in the recovery room as the primary outcome to check the effect of the ESP block. We added these comments in the method section.

       “Since the previous study reported that the intensity of postoperative pain was the most severe in the recovery room [14], we thought that the control of pain in the recovery room is very important. Thus, we set postoperative pain score in the recovery room as the primary outcome to check the effect of the ESP block.”

Point 3: Results – Page 4 lines 121-123: In the results section where the authors describe their primary results the sentence is difficult to follow. The results are listed in the text for multiple parameters sequentially; however, the corresponding patient groups are not listed until after the data. Thus it is unclear which data point refers to which group until reaching the end of the sentence. The sentence should be re-structured so the reader can more easily follow these important and primary results of the study. List the two groups before the data. Ie. “In ESPB group vs control, NRS at rest (4 vs 5) …”etc

Thank you for your advice. And we revised the sentences in the methods section.

        “In the ESPB group vs the control, NRS at rest in the recovery room (4 [3–5] vs. 5 [4–7], p = 0.009), at 6 h after surgery (2 [1–3] vs. 3 [2–4], p = 0.003) and at 24 h after surgery (2 [1–2] vs. 2 [2–3], p = 0.002), were significantly low. In the ESPB group vs the control, NRS on cough in the recovery room (5 [4–6] vs. 6 [5–8], p = 0.001) and at 6 h after surgery (3 [3–4] vs. 4 [3–5], p = 0.040) were significantly low, while the NRS on cough at 24 h after surgery did not differ between the groups (3 [3–4] vs. 3 [2–4], p = 0.158).”

Point 4: For the recovery room timepoint (Table 2) Clarify if “total dose of fentanyl rescue” (used in text) and “Total administered dose of fentanyl” (used in table) are only bolus doses vs including the basal rate. Use a consistent terminology between these two terms uses in text and in the figure..

Thank you for your advice. We revised the sentences.

“In the recovery, the total administered dose of fentanyl bolus (50 [50–100] μg vs. 100 [50–100] μg, p = 0.032) were significantly lower in the ESPB group than in the control group.”

Point 5: Since Table 2 and 3 are reporting similar data at separate time points they could be grouped into one table. If this was decided to keep recovery room separate from other time points the importance of this may be clarified by the authors response to point 2.

Thank you for your advice. We revised Table 2 and 3, and added the figures with pain scores in the result section.

Point 6: In 6 hour and 24 hour timepoint, (Table 3) – Please clarify again whether cumulative fentanyl dose includes basal infusion and boluses.

Thank you for your advice. And we added the description in Table 2.

“PCA: patient-controlled analgesia including fentanyl (800 μg) in normal saline (100 ml)(basal infusion rate of 2 ml/h, intermittent bolus of 0.5 ml with a lock-out in-terval of 15 min).”

Point 7: Since medians are reported for NRS it makes it difficult to appreciate the difference in the groups. For 24 hour timepoint median is reported as 2 for both groups with a significant difference of p= 0.002. The authors might consider a visual representation of the difference between the groups for the most important time points. For example a dot plot where you can see the distribution of pain scores for the ESP block group vs control? This may add a nice visual to support the benefit and could also display the change over the first 24 hours for each group if done for recovery room, 6 hour and 24 hours time points.

Thank you for your advice. And we added the figure of pain scores (box and whisker plot) in results sections

Reviewer 3 Report

The title correctly reflects the subject of the article. The abstract correctly describes the manuscript. The introduction partially reflects the problem of pain management after laparoscopic colorectal surgery.

Pain after laparoscopic colorectal surgery originates from two sources.  First, from the abdominal wall incision and second, from visceral dissection. The regional blocks including the ESPB block improves the parietal pain but not the visceral pain (Kahokehr et al. Ann Surg2011 Jul;254(1):28-38). Intraperitoneal instillation of ropivacaine has shown great efficacy in the treatment of pain, opioid consumption and postoperative recovery in several abdominal surgeries including colorectal surgery (Duffield et al. Dis Colon Rectum 2018). 

The pain scores in the intervention group, despite the significant statistical difference, remains high (NRS 5 vs. 4 at rest and 6 vs. 5 during coughing) and this is most likely due to the visceral pain that remains undertreated.       

The article by Lindberg et al. describes significant postoperative pain in laparoscopic colorectal surgery compared to a laparotomy approach, we find it hard to believe of what we observe in our daily practice. As described in your first line of introduction, several studies have clearly shown the beneficial effect of laparoscopy compared to laparotomy on painn treatement and postoperative recovery.  

Author Response

Dear reviewer #3,

Thank you for reviewing the manuscript.  Answers to your suggestions are written below and changed parts of the manuscript are written in red characters.

Response to Reviewer 3 Comments

Point 1: The title correctly reflects the subject of the article. The abstract correctly describes the manuscript. The introduction partially reflects the problem of pain management after laparoscopic colorectal surgery.

Pain after laparoscopic colorectal surgery originates from two sources.  First, from the abdominal wall incision and second, from visceral dissection. The regional blocks including the ESPB block improves the parietal pain but not the visceral pain (Kahokehr et al. Ann Surg. 2011 Jul;254(1):28-38). Intraperitoneal instillation of ropivacaine has shown great efficacy in the treatment of pain, opioid consumption and postoperative recovery in several abdominal surgeries including colorectal surgery (Duffield et al. Dis Colon Rectum 2018).

The pain scores in the intervention group, despite the significant statistical difference, remains high (NRS 5 vs. 4 at rest and 6 vs. 5 during coughing) and this is most likely due to the visceral pain that remains undertreated.      

Thank you for your advice. We agree with you. And we added the comments about limitation of ESPB in the discussion section.

“Pain after laparoscopic colorectal surgery originates from the abdominal wall inci-sion and visceral dissection. However, the regional blocks including the ESP block may improve the parietal pain but not the visceral pain [23]. Intraperitoneal instillation of ropivacaine has shown great efficacy in the treatment of pain, opioid consumption and postoperative recovery in several abdominal surgeries including colorectal surgery [24]. In this study, the pain scores of ESPB group, despite the significant statistical difference, re-mained high (median NRS at rest was 4 and NRS on cough was 5 in the recovery room). This result is most likely due to the visceral pain that remains undertreated. “

Point 2: The article by Lindberg et al. describes significant postoperative pain in laparoscopic colorectal surgery compared to a laparotomy approach, we find it hard to believe of what we observe in our daily practice. As described in your first line of introduction, several studies have clearly shown the beneficial effect of laparoscopy compared to laparotomy on pain treatement and postoperative recovery.

We agree with you. As you pointed out, several studies have clearly shown that laparoscopic cololectal surgery is less painful than the conventional laparotomy approach. Thus, we revised these sentences about the benefit of laparoscopic surgery in the introduction and discussion sections.

“In general, laparoscopic colorectal surgery is less painful than open laparotomy. But, an earlier report has shown that laparoscopic surgery could be more painful on the first postoperative day [5].

 “A previous study has shown that laparoscopic surgery could show high intensity of pain and high requirements of analgesics immediately after surgery [5].”

Round 2

Reviewer 1 Report

The authors did not explained clearly that why many patients had the NRS more than 4, which  was attributed to the unsuccessful analgesic regimen in this study.  It reflected that the authors' work were not well performed.